# Evaluation of Poly(*N*-Ethyl Pyrrolidine Methacrylamide) (EPA) and Derivatives as Polymeric Vehicles for miRNA Delivery to Neural Cells

**DOI:** 10.3390/pharmaceutics15051451

**Published:** 2023-05-10

**Authors:** Altea Soto, Manuel Nieto-Díaz, Enrique Martínez-Campos, Ana Noalles-Dols, María Asunción Barreda-Manso, Felipe Reviriego, Helmut Reinecke, David Reigada, Teresa Muñoz-Galdeano, Irene Novillo, Alberto Gallardo, Juan Rodríguez-Hernández, Ramón Eritja, Anna Aviñó, Carlos Elvira, Rodrigo M. Maza

**Affiliations:** 1Molecular Neuroprotection Group, Hospital Nacional de Parapléjicos (SESCAM), 45071 Toledo, Spain; mnietod@sescam.jccm.es (M.N.-D.); ana.noalles.dols@gmail.com (A.N.-D.); dreigada@sescam.jccm.es (D.R.); tmunozd@sescam.jccm.es (T.M.-G.); inovilloa@externas.sescam.jccm.es (I.N.); 2Polymer Functionalization Group, Instituto de Ciencia y Tecnología de Polímeros-Consejo Superior de Investigaciones Científicas (ICTP-CSIC), Departamento de Química Macromolecular Aplicada, Juan de la Cierva 3, 28006 Madrid, Spain; e.martinez.campos@csic.es (E.M.-C.); freviriegop@ictp.csic.es (F.R.); hreinecke@ictp.csic.es (H.R.); gallardo@ictp.csic.es (A.G.); jrodriguez@ictp.csic.es (J.R.-H.); celvira@ictp.csic.es (C.E.); 3Group of Organic Synthesis and Bioevaluation, Associated Unit to the ICTP-IQM-CSIC, Instituto Pluridisciplinar, Universidad Complutense de Madrid, Paseo Juan XXIII, n^◦^ 1, 28040 Madrid, Spain; 4Functional Exploration and Neuromodulation of the Central Nervous System Group, Hospital Nacional de Parapléjicos (SESCAM), 45071 Toledo, Spain; mbarreda@sescam.jccm.es; 5Department of Surfactants and Nanobiotechnology, Institute for Advanced Chemistry of Catalonia (IQAC), Spanish National Research Council (CSIC), Jordi Girona 18-26, 08034 Barcelona, Spain; recgma@cid.csic.es (R.E.); aaagma@cid.csic.es (A.A.); 6Networking Center on Bioengineering, Biomaterials and Nanomedicine (CIBER-BBN), 08034 Barcelona, Spain

**Keywords:** poly (*N*-ethyl pyrrolidine methacrylamide), polymeric delivery systems, miRNA transfection, neural cells, neurons, in vitro analyses

## Abstract

MicroRNAs (miRNAs) are endogenous, short RNA oligonucleotides that regulate the expression of hundreds of proteins to control cells’ function in physiological and pathological conditions. miRNA therapeutics are highly specific, reducing the toxicity associated with off-target effects, and require low doses to achieve therapeutic effects. Despite their potential, applying miRNA-based therapies is limited by difficulties in delivery due to their poor stability, fast clearance, poor efficiency, and off-target effects. To overcome these challenges, polymeric vehicles have attracted a lot of attention due to their ease of production with low costs, large payload, safety profiles, and minimal induction of the immune response. Poly(*N*-ethyl pyrrolidine methacrylamide) (EPA) copolymers have shown optimal DNA transfection efficiencies in fibroblasts. The present study aims to evaluate the potential of EPA polymers as miRNA carriers for neural cell lines and primary neuron cultures when they are copolymerized with different compounds. To achieve this aim, we synthesized and characterized different copolymers and evaluated their miRNA condensation ability, size, charge, cytotoxicity, cell binding and internalization ability, and endosomal escape capacity. Finally, we evaluated their miRNA transfection capability and efficacy in Neuro-2a cells and rat primary hippocampal neurons. The results indicate that EPA and its copolymers, incorporating β-cyclodextrins with or without polyethylene glycol acrylate derivatives, can be promising vehicles for miRNA administration to neural cells when all experiments on Neuro-2a cells and primary hippocampal neurons are considered together.

## 1. Introduction

Diseases of the central nervous system (CNS) persist as some of humanity’s most challenging conditions, both because of the deterioration of qualities we closely associate with being human and because in most cases, they lack effective therapies. Nucleic acid-based therapies have emerged as powerful therapeutic approaches for the treatment of CNS injuries and neurological disorders by delivering genes, sequence regulatory molecules, oligonucleotides, or even genetically modified cells. Specifically, RNA therapeutics have some advantages, such as the high specificity towards pathogenic targets, decreasing toxicity associated with off-target effects, and relatively low-dose requirement to achieve therapeutic effects [1,2,3,4]. Among them, this study focuses on microRNA (miRNA)-based therapies. MiRNAs are endogenous 21–23-long oligonucleotides that regulate the expression of hundreds of proteins to control cells’ function in physiological and pathological conditions. miRNA-based therapeutics act at the post-transcriptional level, and therefore, they can downregulate any disease-causing proteins. Despite their advantages, miRNA therapeutics suffer from delivery limitations caused by serum instability, fast clearance, immunogenicity, low cellular uptake, and inability to penetrate the blood–brain barrier (BBB) of miRNA molecules, which hinder its clinical application in the CNS [3,5,6]. To overcome these challenges, different nucleic acid carriers have been examined. Polymeric delivery systems have attracted a lot of attention due to their merits, such as the ease of production within low costs, great customizability, large payload, and reduced induction of the immune response, among others. However, most of them have low transfection efficiencies and can be cytotoxic, which severely limits their utility in neuron-targeted delivery applications. Polymeric vehicles are typically made of cationic monomers containing primary, secondary, and/or tertiary amine groups that can condense negatively charged nucleic acids via entropically driven electrostatic interactions, forming complexes named polyplexes [7,8,9]. Polymeric vehicles have highly versatile chemistries and physical characteristics; so, they can be functionalized to provide the controlled release of cargo, support transfection, and affect biodistribution, among others [2,9,10,11]. Over the years, diverse polymers have been studied as nucleic acids carriers, such as poly (ethylenimine) (PEI), poly (lactide-co-glycolide) (PLGA), chitosan (CS), poly (L-lysine) (PLL), and poly (*N*-ethyl pyrrolidine methacrylamide) (EPA), among others [9,10,12,13,14,15]. However, toxicity and low transfection efficiency issues have hindered their clinical application as homopolymers. Copolymerization could play an important role in increasing cytocompatibility and transfection efficiency [16]. Specifically, EPA showed good DNA transfection efficiencies in fibroblasts cell lines in vitro when it was copolymerized with *N*,*N*’-dimethylacrylamide (DMA) [17], 1-vinylimidazole (VI) [18], *N*-hydroxypropyl methacrylamide (HPMA) [16], ß-cyclodextrins (CDs), and hydroxylated (GLCSt) and permethylated (MeGLCSt) forms of ɑ-glucose [19]. Nevertheless, none of these copolymers have been tested in studies on the miRNA transfection of neural cells.

The present study aims to evaluate the potential of poly (*N*-ethyl pyrrolidine methacrylamide) (poly-EPA) and its copolymers with HPMA, glucose, CDs, and poly(ethylene glycol) methyl ether acrylate (APEG) as vehicles for the delivery of RNAi therapeutics to neural cells. For these purposes, we synthesized and characterized linear homopolymers, copolymers, and terpolymers and evaluated their properties for miRNA delivery in cultures of Neuro-2a cells and primary hippocampal neurons, two standard in vitro models. The study was designed as a sequence of screening steps in which only those polymers displaying optimal complexation and protective properties were included in the subsequent analyses of toxicity and cell binding, and only the non-toxic and efficient ones were fully characterized (complex size, charge, and morphology, endosomal escape capacity, and transfection efficiency). Throughout these analyses, we employed two different miRNAs as the cargo, miR-138a-5p and cel-miR-67. The first one is a conserved mammalian miRNA that is highly expressed in the CNS and has neuroprotective properties and may constitute a therapeutic tool for neurodegenerative processes in pathologies such as spinal cord injuries [20], whereas cel-miR-67 is a *C. elegans* miRNA [21] without known targets in mammalian cells, and it was included as a control.

## 2. Materials and Methods

### 2.1. Synthesis of Homopolymers, Copolymers, and Terpolymers

The synthesis and characterization of respective synthetic monomers, homopolymer poly-EPA (EPA) [17], and copolymers, EPA-GLCSt (EG) and EPA-MGLCSt (EM) [19], EPA-HCD (ECD) [22], and EPA-HPMA (EHPMA) [16], have been previously described. On the other hand, EPA-HCD-APEG (ECDP-) terpolymers were prepared via free radical polymerization procedures where the monomers and the initiator (AIBN, 2,2′-azobisisobutyronitrile, Fluka) were dissolved in *N*,*N*’dimethyl formamide (DMF) at concentrations of 1.0 and 1.5 × 10^−2^ M, respectively, using 0.85/0.1/0.05 (ECDP5), 0.8/0.1/0.1 (ECDP10), and 0.7/0.1/0.2 (ECDP20) molar fractions of EPA/HCD/APEG, respectively. The radical initiator, AIBN, was recrystallized in methanol. The monomer poly (ethylene glycol) methyl ether acrylate (APEG), Mn 480 (Sigma-Aldrich, St. Louis, MO, USA), was used without further purification. Gaseous N_2_ was flushed through the solutions for 20 min. Polymerization reactions were carried out at 60 °C for 24 h to attain total conversion. Poly-EPA homopolymer (EPA) was also synthesized under identical conditions. All polymers were dialyzed against water using Spectra/Por (Spectrum Laboratories Inc., Rancho Dominguez, CA, USA) dialysis membranes, with a molecular weight cut-off 3.5 kDa, and then freeze dried. The composition and chemical structures of the different monomers, copolymers and terpolymers are schematized in Figure 1.

The gold standard polymeric carrier, 25-kDa hyperbranched polyethyleneimine (PEI25K, MW = 25,000, d/PDI = 2.5) (Sigma Aldrich, Milwaukee, WI, USA, cat#408727), was used as the reference. 

### 2.2. miRNA Mimics

Throughout this study, we employed the following nucleic acids: (i)A synthetic miR-138 mimic (miR-138 mimic), a duplex RNA derivative composed of an antisense, and a sense strand. The antisense strand is a 5′-phosphate (P) modified RNA strand (5′-P-AGCUGGUGUUGUGAAUCAGGCCG-3′) corresponding to the human miR-138-5p sequence obtained from commercial sources (Merck-Sigma-Aldrich España, Madrid, Spain). The sense RNA strand was designed as the complementary sequence (5′-CGGCCUGAUUCACAACACCAGCU-3′) and was synthesized at a 1 μmol scale using 2’-O-TBDMS (*t*-butyldimethylsilyl) protected phosphoramidites (A, G, C and U) using a 3400 ABI DNA/RNA synthesizer. Synthesis was performed following standard protocols on DMT-ON mode. After the ammonia treatment, the solvent was evaporated to dryness, and 0.15 mL of triethylamine hydrofluoride/triethylamine /*N*-methylpyrrolidone (4:3:6) was added to remove *t*-butyldimethylsilyl groups. TBDMS-protected RNAs were incubated 2.5 h at 65 °C as described by Wincott et al. [23]. The deprotection reaction was quenched, and the crude product was purified using OPC cartridges (Glen Research). Oligoribonucleotides were quantified via UV absorption and characterized via MALDI-TOF mass spectrometry. The miR-138 mimic was obtained by annealing equimolar amounts of sense and antisense strands in 100 μL of 10 mM TRIS and 50 mM NaCl buffer. The resulting solution was heated at 94°C for 2 min, and then allowed to cool until it reached room temperature (RT). Then, 3 M sodium acetate pH = 5.2 was added (10 μL) along with EtOH (96%) (275 μL). Samples were stirred and precipitated at −20 °C. Finally, samples were centrifuged at 4 °C (15 min, 12000 rpm), and the supernatant was removed. The resulting pellets were dried using N_2_.(ii)A commercial cel-miR-67 mimic with a minimal amount of shared sequence identity with vertebrate miRNAs and no identified effects in vertebrate cells (miRIDIAN miRNA mimic negative control#1; cat#CN-001000-01; Dharmacon^TM^) was employed as a negative control for functional assays (Neg. Ctrl mimic).(iii)Commercial Cy3-labeled cel-miR-67 (miRIDIAN miRNA Mimic Red Transfection Control; cat#CP-004500-01-05; Dharmacon^TM^) was employed as a negative control in flow cytometry and immunofluorescence experiments (Cy3 Neg. Ctrl mimic).(iv)A DNA oligonucleotide with sequence equivalence to miR-138 mimic was employed for DLS and ζ-potential analyses to reduce the costs and degradation issues.

### 2.3. Characterization of Polymer Systems

EPA-HCD-APEG (ECDP-) terpolymers were characterized using the following techniques: (i)Spectroscopic techniques: Polymer systems were characterized via ^1^H nuclear magnetic resonance spectroscopy (^1^H-NMR). Spectra were recorded in 5% deuterated chloroform (CDCl_3_) solution, deuterated methanol (CD_3_OD), or deuterated DMSO (DMSO-d_6_) with a Varian XLR-300 spectrometer using trimethylsilane (TMS) as the internal standard.(ii)Chromatographic techniques: The number average molecular weight (Mn) and polydispersity index (Ð) of the polymers were measured via gel permeation chromatography (GPC) with a Perkin Elmer (Waltham, MA, USA) chromatographic system equipped with a Waters (Milford, MA, USA) model 2414 refractive index detector using Styragel (300 mm × 7.8 mm and 5 μm nominal particle size) HR3 and HR5 water columns. Dimethylformamide (DMF) with 1 wt% LiBr was used as an eluent. Measurements were performed at 70 °C at a flow rate of 0.7 mL/min using a polymer concentration of 4 mg/mL. Calibration was performed with monodispersed polystyrene standards in the range between 2.0 and 9000.0 kDa.

### 2.4. Polyplexes Formation 

Polymers condense miRNA via ionic interactions of the amine groups (N) of the polymer and the phosphate groups (P) of nucleic acid. Various formulations of polymer/RNA complexes (polyplexes) were prepared in distilled water or Tris-EDTA solution (TE, pH = 7), depending on the assay, with N/P molar ratios ranging from 1/1 to 100/1. The N/P ratios of polyplexes of PEI25K did not match the ratios of the other polymeric systems in the analyses because they were originally analysed as mass/mass ratios and only later transformed into N/P ratios (N/P ratio = 7.5 corresponds to a mass/mass ratio equal to 1). For polyplex formation, RNA and polymer solutions were combined and left to stand for 20 min at RT.

### 2.5. Condensation Efficiency Assays 

To determine the miRNA condensation efficiency of different polymers, a SybrGold assay based on Xie et al. [24] was employed. Sybr Gold is a dye that intercalates into nucleic acids and produces fluorescence. When RNA is condensed by polymers, it becomes inaccessible to SYBR Gold, and the level of fluorescence decreases. In this assay, polyplexes at different N/P ratios (from 1/1 to 100/1) were formed by combining 50 nM of miR-138 mimic with the different polymers in Tris-EDTA solution (TE) (pH = 7), and then transferred to a black, flat, 96-wells plate. Subsequently, 25 µL of 5× SybrGold nucleic acid stain (ThermoFisher Scientific, Waltham, MA, USA) was added and incubated at RT for 10 min in the dark. PEI25K polyplexes at different N/P ratios (from 2/1 to 75/1) were included as a reference. The fluorescence signal was evaluated by exciting the sample at 495 nm and measuring the emission at 550 nm using a fluorescence plate reader (Infinite M200, Tecan Group LTD. Mannendorf, Switzerland). Samples of uncomplexed miRNA (N/P ratio of 0/1) were prepared, treated, and measured in the same way as the polyplex samples were, and their values were used as baseline for 100% free miRNA. Free RNA in each condition was measured in triplicates in at least three independent experiments.

### 2.6. RNA Protection Assays

In order to evaluate the protection conferred by polymers to RNA once bound, RNA degradation was measured after polyplex incubation with RNases following the procedure described by Nakanishi et al. [25] with minor modifications. Briefly, polyplexes at different N/P ratios according to their maximum condensation efficacy established in the condensation assays (5/1 for EPA, EHPMA, ECD, ECDP5, and ECDP10; 10/1 for ECDP20 and EMCD; 20/1 for G10 and M10) were formed by combining 1 µL of 50 nM of miR-138 mimic with the corresponding amounts of the different polymers in the Tris-EDTA solution (TE) (pH = 7). PEI25K polyplexes at an N/P ratio of 15/1 were included as a reference. A total of 5.5 µL of the polyplexes solution was incubated with 1 µL of 0.015 mg/mL RNase A (ThermoFisher Scientific) for 60 min at 37 °C. Then, RNase A was blocked using 1.5 µL of RNasin inhibitor (Applied Biosystems, Foster City, CA, USA), and miR-138 mimic was released from the polyplexes via incubation with 1.5 µL of 1000U ds/mL heparin (Ramon Sala Laboratories, Barcelona, Spain) for 30 min at RT. Finally, the resulting solution was mixed with 10 µL of loading buffer (95% formamide (Sigma-Aldrich), 4.97% glycerol (Sigma-Aldrich), and 0.03% bromophenol blue (Affymetrix/USBTM)), and the samples were analyzed via electrophoresis in a 40% polyacrylamide (acrylamide:bis-acrylamide = 19:1) gel with a 42% of urea (Sigma-Aldrich) prepared in TBE buffer (100 mM Tris (Sigma-Aldrich), 100 mM Boric Acid (Sigma-Aldrich), and 2 mM Na_2_EDTA (Sigma-Aldrich) at pH = 8.2). Electrophoresis was carried out with 1× TBE at a constant voltage of 150 V for 90–120 min. A dsRNA-ladder (cat#: N0363S; New England BioLabs) formed by a set of 7 annealed double-stranded RNA molecules was used. RNA was visualized under a UV transilluminator after staining for 15 min with SybrGold. Samples of 10 pmol of uncomplexed miRNA mimic with and without treatment with RNase were included as negative (None−) and positive (None+) controls, respectively.

### 2.7. Cell Culture 

To evaluate the toxicity, efficacy, and specificity of different polymers for the cell transfection of miRNA mimics, we used Neuro-2a mouse neuroblastoma (cat#: CCL-131, ATCC; RRID#CVCL_0470) and C6 rat brain glioma (cat#: CCL-107, ATCC; RRID##CVCL_0194) cell lines, as well as a primary culture of embryonic neurons from rat hippocampi. Neuro-2a and C6 cell lines were cultured in Dulbecco’s modified Eagle’s medium (DMEM; Gibco, New York, NY, USA) supplemented with 10% fetal bovine serum (FBS; Gibco), 1% penicillin/streptomycin solution (P/S; Gibco), and 1% glutaMAX (Gibco) at 37 °C in a humidified incubator containing 5% CO_2_. Primary hippocampal neurons were obtained from 17–18-day-old (E17–18) Wistar rat embryos. Briefly, after dissection, hippocampi were subjected to enzymatic digestion in Hanks′ Balanced Salt Solution (HBSS) medium without calcium and magnesium (Hyclone, GE Healthcare) supplemented with Trypsin (1×; Thermo Fisher Scientific) and DNase (20 mg/mL; Roche) for 15 min at 37 °C. Trypsin and DNase were washed out with HBSS with calcium and magnesium (Hyclone) and mechanically disrupted by passing the tissue sample several times through a glass pipette in Minimum Essential Medium (MEM; Gibco) supplemented with 10% horse serum (HS; Fisher Scientific). The obtained cell suspension was added in poly-L-lysine pre-coated multi-well culture plates for at least 4 h at 37 °C in a humidified incubator containing 5% CO_2_. Finally, the culture medium was changed to Neurobasal Medium (Gibco) supplemented with 2% B-27 Supplement (Gibco), 1% glutaMAX, and 1% P/S, and cultures were incubated at 37 °C in a humidified incubator containing 5% CO_2_ for 2 days before use.

### 2.8. Cell Transfection

Cells were transfected with different N/P ratios of polymer-miRNA polyplexes. Depending on the assay, the miRNAs previously described were used at a final concentration of 50 nM: (i) miR-138 mimic; (ii) Neg. Ctrl mimic, and (iii) Cy3 Neg. Ctrl mimic. In all experiments, PEI25K polyplexes at different N/P ratios were included as references. 

### 2.9. Flow Cytometry 

Flow cytometry was employed for the simultaneous analysis of cell binding and toxicity caused by different polyplexes. Fluorescent-labeled miRNA (Cy3-labeled cel-miR-67 negative control mimic) (Cy3 Neg. Ctrl mimic; Dharmacon^TM^) was used for cell binding analyses. Briefly, 40,000 Neuro-2a cells/well were seeded in 48-well plates. Twenty-four hours later, 25 μL of the correspondent polyplexes prepared with Cy3 Neg. Ctrl mimic at a final concentration of 50nM and different amounts of polymer at appropriate N/P ratios (3/1 and 10/1 for EPA, ECD, ECDP5, and ECDP10; 3/1, 10/1, and 30/1 for EMCD and EHPMA; 1/1, 10/1, and 25/1 for G10 and M10) were added to the cell culture. PEI25K polyplexes at N/P ratios of 7.5/1, 75/1, and 187/1 were included as references. After 24 h, cell death was analyzed by using the apoptosis detection kit Annexin V FITC (Immunostep) and SYTOX (Invitrogen, ThermoFisher Scientific) and employing an FACS Canto II flow cytometer (BD Biosciences, Franklin Lakes, NJ, USA). Cy3-labeling was conducted for the detection of cells presenting Cy3 Neg. Ctrl mimic (miR+ cells). A dot plot showing the pulse width versus area was used to distinguish between single cells and aggregates, and at least 5000 gated single events were analyzed for every condition by the FACS Diva 6.1 (BD Biosciences) and the Flow Jo software (Celeza GmbH, Olten, Switzerland). All experiments were performed in duplicate in at least three independent experiments.

### 2.10. Cell Viability Assays 

The effect of different polyplexes on cell viability was measured via the (4,5-dimethylthiazol-2-yl)-2,5-diphenyltetrazolium bromide (MTT) assay. Briefly, 10,000 Neuro-2a cells per well were seeded in 96-well plates. Twenty-four hours later, cells were incubated with polyplexes at N/P ratios of 3/1 and 10/1 for 24 h at 37 °C and 5% CO_2_ in a cell incubator. PEI25K polyplexes at N/P ratios of 7.5/1 and 75/1 were included as references. Then, cells were incubated with 10 µL MTT reagent (5 mg/mL; Sigma-Aldrich) for 1–2 h at 37 °C and solubilized with 100 µL of solubilization reagent (hydrochloric acid (HCl) 0.1 M in isopropanol) at RT for 20 min. Finally, absorbance at 570 and 660 nm was determined using a plate reader luminometer (Infinite M200, Tecan Group LTD. Mannendorf, Switzerland). All analyses were performed in triplicates in at least three independent experiments.

### 2.11. Particle Size and Zeta Potential

Various formulations of polymer/DNA complexes were prepared by combining 50 nM of DNA oligonucleotide with the sequence equivalence to miR-138 mimic (described in Section 2) and different polymers in UltraPureTM Distilled Water (Invitrogen) at an N/P ratio of 5/1. PEI25K polyplexes at an N/P ratio of 7.5/1 were included as references. Polyplexes were analyzed after 20 min of incubation at RT.

The determination of the particle sizes of complexes was carried out at 25 °C using a size analyzer (N5 Submicron Particle Size Analyzer, Beckman Coulter) and Dynamic Light Scattering (DLS) (Nano-ZS, Malvern instrument). The surface charge of the complexes (ζ-potential) was assessed using a Zetamaster system (Malvern Instruments). Every measurement was carried out three times. Data acquisition was performed with ALVCorrelator Control software, and the counting time varied for each sample from 300 s up to 600 s.

### 2.12. Immunofluorescence and Confocal Analysis

In order to evaluate the capacity of different polymers to specifically transfect miRNAs into Neuro-2a cells and primary neuron cultures, 40,000 cells/well were seeded over poly-L-lysine pre-coated 12 mm round coverslips inside 24-well plates. Twenty-four hours later, cells were transfected with polyplexes prepared via the combination of 50 nM Cy3 Neg. Ctrl mimic and EPA, EHPMA, ECD, ECDP5, and ECDP10 polymers at an N/P ratio of 5/1. PEI25K polyplexes at an N/P ratio of 7.5/1 were included as references. After 24 h, cells were fixed with 4% paraformaldehyde (PFA; Sigma-Aldrich) for 30 min at RT and washed with 1× Phosphate-buffered Saline (PBS) for us to perform the immunofluorescence assay. Briefly, fixed cells were permeabilized and blocked with 0.2% Triton X-100 and 3% Bovine serum albumin protein (BSA), respectively, in PBS for 30 min at RT. Primary neuron cultures were immunostained for 2 h at RT with a neuronal-specific marker, mouse anti-β−tubulin III isoform antibody (1:500, Millipore, cat#MAB1637, RRID: AB_2210524), followed by a 2 h incubation at RT with a goat anti-mouse Alexa Fluor 633 conjugated secondary antibody (1:500, Molecular Probes, cat# A-21050, RRID: AB_141431). Neuro-2a cells were immunostained for 2 h at RT with a mouse anti-ɑ-tubulin isoform antibody (1:1000; Sigma-Aldrich, cat#: T6074, RRID: AB_477582), which was followed a 2 h incubation at RT with a goat anti-mouse Alexa Fluor 633 conjugated secondary antibody (1:500, Molecular Probes, cat#A-21050, RRID: AB_141431). Then, both Neuro-2a and neurons nuclei were stained with DAPI (4’,6-diaminidine-2-phenylindole, 1:10,000, Merck, cat#D9542) for 5 min at RT and mounted on Lab Vision™ PermaFluor™ Aqueous Mounting Medium (Thermo Fisher Scientific, cat#TA-030-FM). Cell cultures were imaged using an epifluorescence microscope (DM5000B, Leica Microsystem GmbH), Wetzlar, Germany) with a 20× objective and analyzed using ImageJ software (National Institutes of Health, Bethesda, MD). The number of miRNA-stained cells was assessed by counting the number of cells that had Cy3 Neg. Ctrl mimic inside at 5 images per well. All experiments were performed in duplicate in at least three independent experiments.

To confirm polyplex internalization, each culture preparation was imaged using a LeicaTCS SP5 fast scan confocal microscope equipped with a PL APO 63×/1.3 objective. Representative images for different polyplexes were obtained in stacks up to 58 Zs, comprising the whole cell height for the region under analysis. For the deconvolution of Cy3 Neg. Ctrl mimic and Alexa 633 (β-III-tubulin) microscopy images, we used the Richardson–Lucy deconvolution algorithm included in the DeconvolutionLab2 plugin for FIJI [26]. PSF models required for the deconvolution process were created in FIJI using Bob Dougherty’s Diffraction PSF 3D plugin. Images were analyzed using the orthogonal views tool of ImageJ software.

### 2.13. Endosomal Escape Assays

In order to evaluate polymer-mediated endosomal escape, we performed a calcein leakage assay described by Kongkatigumjorn et al. [27], which measures the escape of calcein from disrupted intracellular vesicles. Briefly, 20,000 C6 cells/well were seeded over 12 mm round coverslips inside 24-well plates. Twenty-four hours later, calcein (in DMSO; #C0875-5G, Sigma-Aldrich) was added to a final concentration of 200 μg/mL and incubated for 2 h at 37 °C with 5% CO_2_. Cells were washed in PBS to remove excess calcein and transfected with 50 nM of cel-miR-67 negative control mimic (Neg. Ctrl mimic) complexed with different polymers at an N/P ratio of 5/1. PEI25K polyplexes at an N/P ratio of 7.5/1 were included as references. Twenty-four hours later, cells were fixed with 4% PFA for 20 min at RT, washed with PBS, and incubated with 5 μg/mL of Hoechst (Sigma-Aldrich) for 15 min at RT. Finally, coverslips were mounted on glass slides with PermaFluorTM Aqueous Mounting Medium (ThermoFisher Scientific). Preparations were imaged with an epifluorescence microscope (DM5000B, Leica Microsystem GmbH, Wetzlar, Germany) using a ×40 objective. Images were processed using ImageJ software, and endosomal escape was assessed using a custom macro (the employed ImageJ macro is available upon request) to count the number of calcein-stained vesicles (0.05–1 μm^2^ size) and the number of cells in a total of 9 images per condition in at least three independent experiments.

### 2.14. Dual-Luciferase Reporter Assay

In order to evaluate the transfection efficacy of different polyplexes, we employed a dual-luciferase reporter assay whose expression is controlled by miRNA of interest. Therefore, the luciferase signal will decay when miRNA is efficiently transfected by the polyplexes. We used a modified pmiRGLO Dual-luciferase mRNA Target Expression Vector (Promega, WI, USA, http://www.addgene.org/vector-database/8236/, accessed on 1 June 2022) whose luciferase expression is controlled by specific binding sites for cel-miR-67 miRNA (pmiRGLO-cel-miR-67). Briefly, we seeded 10,000 Neuro-2a cells/well in 96-well plates to be transfected 24 h after with 200 ng/well of pmiRGLO-cel-miR-67 using TurboFECT^TM^ Transfection Reagent (Fisher Scientific). A second round of transfection was performed 24 h later with 50 nM of cel-miR-67 negative control mimic (Neg. Ctrl mimic) complexed with the different polymers at defined N/P ratios. PEI25K polyplexes at an N/P ratio of 7.5/1 were included as references. The reporter gene expression was evaluated by measuring firefly and Renilla luciferase activities using the Dual-GLO luciferase assay system (Promega) in an Infinite M200 plate reader (Tecan) according to the manufacturer’s protocol. Firefly emission data were normalized to Renilla load control levels and expressed as the firefly/Renilla ratio. Luminescence values of only pmiRGLO-cel-miR-67 transfected cells (without miRNA) were used as control. All experiments were performed in triplicate in at least three independent experiments.

### 2.15. Data Analysis

All data are expressed as mean ± SEM or SD, as indicated in the figure legends. For statistical analysis, data were standardized to account for baseline variations among experiments. One-tailed or two-tailed *t*-tests were applied in all cases according to the expected outcomes. Statistical analyses were carried out and graphics were produced using https://www.socscistatistics.com/tests/ (accessed on 20 January 2023) and Prism Software 5 (GraphPad Software Inc., La Jolla, CA, USA). Differences were considered to be statistically significant for *p* values < 0.05.

## 3. Results and Discussion

### 3.1. Synthesis and Characterization of Terpolymers

Ten different polymer systems have been prepared to be evaluated as non-viral vectors of miRNAs in terms of transfection capability and efficiency in Neuro-2a cells and primary hippocampal neurons. Poly(N-ethyl pyrrolidine methacrylamide) (EPA) is a cationic, linear, non-toxic, and non-crosslinked polymer that was synthesized and characterized for the first time by Velasco and colleagues in 2007 [15]. The synthesis and characterization of other polymer systems, except terpolymers, ECDP, composed of EPA-HCD-APEG, have been previously described [17,19,22]. EG is a copolymer of EPA with the monomer, GLSt, which is derived from glucose, with 0.9/0.1 (EG10) and 0.95/0.05 (EG5) molar fractions, whereas EM is a copolymer of EPA and MGLSt, a permethylated glucose monomer, with 0.9/0.1 (EM10) and 0.95/0.05 (EM5) molar fractions [19]. The glucose bioactivity of various glucose-conjugated polymers has been described to favor cellular uptake efficiency and improve transfection capabilities in fibroblast cells lines [18], and they may function as probes for tracking cell and tissue uptake and the metabolism of glucose in clinical research [28]. 

EHPMA copolymers are based on EPA and neutral N-2-(hydroxypropyl) methacrylamide (HPMA) prepared with 0.5/0.5 molar fractions, and this has been previously described as the best composition for transfection and biocompatibility [16]. HPMA is a precursor with a biocompatible nature, whose copolymerization of HPMA with cationic precursors enhances the performance as gene carriers of cationic component homopolymers and allows the modulation of the charge density of polymeric entities and their interaction with the surrounding media in a flexible and tailored way [18]. ECD is a copolymer of EPA and a cyclodextrin (HCD)-derivative monomer designed to improve the cellular uptake of polyplexes, and good interactions with the cellular membrane have been described before, and their transfection capability improves when fibroblasts cell lines are used [22]. Due to their unique structure with multiple hydroxyl groups, amphiphilic cyclodextrin-based polymers provide low toxicity, good bioavailability, and biodegradability, as well as the ability to form well-defined aggregated nanostructures [29].

In addition, the good properties of ECDs copolymers observed during the course of initial experiments and the synergistic effects of cyclodextrin and cross-linking monomers [29] led us to prepare ECDP–terpolymers of EPA-HCD-APEG, incorporating the acrylate derivative of PEG to improve their biocompatibility. In this sense, three different molar fraction compositions of EPA/HCD/APEG, 0.85/0.1/0.05 (ECDP5), 0.8/0.1/0.1 (ECDP10), and 0.7/0.1/0.2 (ECDP20), respectively, were prepared. Via ^1^H NMR Spectroscopy, we have determined the molar fractions of the respective monomer in the prepared terpolymer, as shown in Figure 1A. As shown in Figure 1B, for these calculations, we used the integration of ^1^H NMR signals corresponding to the triazole H of CDH at 8.11 ppm, to the protons from the -CH_2_-O-CO- group of APEG at 4.14 ppm, and to the methyl group of EPA at 0.94 ppm. It can be observed Figure 1A) that all monomer molar fractions are similar in the feed and in the prepared terpolymers, with number average molecular weights, Mn, between 17–20 kDa and polydispersities higher than 1.3 in all cases, which are close to those reported for EPA-HCD copolymers [22].

### 3.2. miRNA Condensation Efficiency

Polymers condense miRNA via ionic interactions between polymer amine groups (N) and nucleic acid phosphate groups (P). We employed a SybrGold assay to determine the miRNA condensation efficiencies of different polymers. The condensation curves in Figure 2 show that the EPA homopolymer produced the highest condensation efficiency of all tested polymer systems, condensing more than 90% of the miRNA mimic at N/P ratios below 2/1, improving upon the DNA plasmid condensations reported previously, which showed good complexation abilities, but not at low N/P ratios (1/1 and 2/1; [19,22]). This can be attributed to the polymer having the highest amount of cationic units available for complexing RNA in comparison to those of the prepared copolymer and terpolymers. PEI25K condensed similar percentage of the miRNA mimic, but at higher N/P ratios above 7.5/1. 

EPA copolymers with glycosylated (EG5 and EG10) and permethylated groups (EM5 and EM10) reduced the condensation efficiency, especially glycosylated copolymers, which condensed a maximum of 60% of the miRNA mimic, although the low molar substitution of EPA polymer amine units by neutral glucosidic and permethylated units has almost no influence in the capability of the EPA polycation to complex the miRNA mimic. A reduction in the efficiency to complex nucleic acids in EG and EM copolymers was also described by Redondo et al. [19], but in contrast to the present results, the reductions were higher in permethylated glucose copolymers than they were in the glycosylated ones. EHPMA copolymers also resulted in reduced condensations (70–80%) of the miRNA mimic, which strongly contrast with the full complexation of DNA plasmids observed previously [16]. These differences in complexation properties between the present experiments and those from Redondo et al. [16], as well as those observed in the other copolymers here studied, are likely the result of differences in the complexed nucleic acid. The rather small size of miRNAs mimics (21 bp) likely limits the stability of purely electrostatic polyelectrolyte complexes compared to that of larger plasmidic DNAs [30]. Moreover, this partial complexation can also be attributed to the environmental pH, where most of the amine groups are deprotonated and surrounded by neutral HPMA units, and thus unable to interact with RNA [16].

The hydroxylated cyclodextrin copolymer (ECD) was as efficient as the EPA homopolymer was in complexing the miRNA mimic, although it required higher N/P ratios (5/1). These results contrast with the increased complexation efficiencies of ECD copolymers observed by Redondo et al. [22], which have been ascribed by the authors to the formation of hydrogen bonds between the hydroxyl groups of CD and N or O from amino, or carboxyl, and phosphate groups of DNA, respectively. On the contrary, in agreement with the observations of Redondo et al. [22], its permethylated form (EMCD) is less efficient, with it being able to condense up to an 80% of the miRNA at much higher N/P ratios. The incorporation of PEG into the ECD copolymer reduced the condensation efficiency as the percentage of APEG is increased, therefore reducing the cationic units available to complex, and increasing the N/P ratio that is required to reach maximum condensation values, as previously observed when PEG was incorporated in poly-L-lysine polymers used to condition DNA condensation, nanoparticle shape, and transcriptional availability [31]. However, at low APEG values (5%; ECDP5), the resulting terpolymer was as efficient as the ECD copolymer was, although it required a lower N/P ratio to reach the maximum condensation values, showing similar behavior to that of the EPA homopolymer.

### 3.3. Protection against RNAses

In order to evaluate the protection conferred by polymers to RNA, once it has been bound, RNA degradation was measured after polyplex incubation with RNases following the procedure described by Nakanishi et al. [25]. To test the protection against RNases provided by the polymers, we complexed the miR-138 mimic with polymer systems at 5/1 or 10/1 N/P ratios to ensure maximum condensation. RNA molecules are extremely sensitive to enzymatic degradation by ribonucleases, as illustrated by the complete disappearance of the miRNA mimic band after 1 h of the RNase A treatment, as shown in Figure 3. When it was complexed with the different polymers and treated with RNAse A (+), the miR-138 mimic was fully protected against RNases, as demonstrated by the electrophoretic bands, which did not differ from those of uncomplexed RNA or complexes treated with RNase-free water (−), except for EM10, which showed a clear smear indicative of miRNA degradation due to its low condensation efficiency, as shown in Figure 1. The absence of any sign of degradation in complexed RNA indicates that all polymers tested completely protect RNA, except for EM10, where the increased hydrophobicity of the polymer by increasing the permeability of glucose monomers could expose miRNA to enzymatic degradation.

On the basis of these results, we discarded those polymers unable to condensate above 75% of the miRNA mimics or unable to protect them from RNAses from further analyses. This filtering scheme affected all copolymers of EPA with permethylated and non-permethylated glucose monomers (EM and EG), as well as the terpolymer of EPA with HCD and APEG of 0.7/0.1/0.2 molar fractions (ECDP20). The remaining polymer systems were tested for cytotoxicity and cell binding at different N/P ratios. We decided also to include one representative of EM and EG polymeric systems that was filtered out to confirm that their properties were not really appropriate for miRNA delivery.

### 3.4. Cytotoxicity and Cell Binding of the Polyplexes

As a preliminary estimation of the transfection efficiency and to characterize the polyplex toxicity at individual cell level, we employed flow cytometry to analyze neural Neuro-2a cells after treatments with polyplexes at low (3/1) and high (10/1) N/P ratios bearing the Cy3-labeled negative control oligonucleotide duplex (Cy3 Neg. Ctrl mimic). Twenty-four hours after treatment, cells were stained with Annexin V FITC and SYTOX markers of cell death. SYTOX dye stains necrotic cells by binding to cellular nucleic acids, but it is impermeant to living and apoptotic cells, whereas annexin V stains apoptotic cells. By combining these dyes with the fluorescent Cy3 Neg. Ctrl mimic, flow cytometry analysis not only allowed us to determine the overall viability of the cultures, but also to characterize the percentage of viable Cy3 Neg. Ctrl mimic positive cells (miR+ cells), which reflect cell binding or the internalization of Cy3 Neg. Ctrl. mimic.

Cy3 Neg. Ctrl. mimic by itself labels, on average, 15% of the cells in culture (*p* < 0.05), and compared to this control, PEI25K polyplexes caused the maximum increase in miRNA staining (above 60% miR+ cells, *p* < 0.01 and *p* < 0.001 at N/P ratios of 7.5/1 and 75/1, respectively). Cyclodextrin derivatives, ECD, ECDP5, and ECDP10, also resulted in significant increases at both high and low N/P ratios (except for ECD, which was only effective at increasing cell staining at N/P ratio of 10/1), although the percentage of viable miR+ cells was clearly lower than it was after PEI25K transfection (below 40% miR+ cells). Among the rest of the polymers, only the treatment with the homopolymer of EPA and high doses of EHPMA resulted in percentages of viable miR+ cells higher than those of the control, although the increase was marginal (up to a 20%). On the contrary, and consistent with their low condensation efficiency (see Figure 2), EM10 and EG10 copolymers did not yield percentages of viable miR+ cells above the controls. Failure in the delivery to the cells was not unexpected in the case of EG10, as the original description [20] already raised doubts about the efficiency of this family of copolymers as nucleic acid carriers, and the present study already revealed its failure to efficiently complex miRNAs. On the contrary, the results with EM copolymers strongly contrast with the results of these authors. According to their luciferase assays, EM copolymers were highly effective in transfecting DNA plasmids to fibroblast cell lines, even improving the results from using PEI25K [19]. The methylation of cyclodextrins in EMCD copolymers also resulted in low numbers of viable miR+ cells, which only reach the levels obtained after the treatment with free miRNA at very high N/P ratios (30/1). Even though the requirement of high N/P ratios is to be expected from poor behavior, as observed in condensation analyses, the low numbers of miR+ cells after transfection with EMCD copolymer is surprising. In the original description of this copolymer, the transfection efficiency was very high and was well above the values of PEI25K and at least as good as that of ECD copolymers [22]. 

Whether these differences are due to the target cells or to the cargo is to be determined, but the reduced efficiency to complex miRNA mimics shown by these polymeric systems suggest that the nature of the cargo should be a major factor. Differences may also arise from the evaluation method used, and it could be argued that cell staining with vehicled miRNA is not a precise proxy for transfection efficiency. However, it provides a reasonable estimation of the maximum number of cells that can be effectively transfected, and given that G10, M10, and EMCD show values that are lower than the number of cells stained using the free miRNA mimic, we discarded them from further analyses that were restricted to EPA, EHPMA, ECD, ECDP5, and ECDP10.

Concerning toxicity (Figure 4A), the administration of Cy3 Neg. Ctrl mimic without a vehicle resulted in the death of almost a third of the cells in the culture (*p* < 0.01), reflecting that miRNAs can trigger neurodegeneration and cause neurotoxicity [32]. Compared to the negative control, all tested vehicles did not increase toxicity, except for marginally at high doses of ECD (N/P ratio of 10/1, *p* < 0.1). ECD toxicity was previously described [22] and could be partially related to its microstructure, which tends to form sequences that are richer in CD-bearing units. 

To further evaluate cytotoxicity, we used the well-established MTT measurement of metabolic activity to analyze Neuro-2a viability after treatments with different polyplexes at low (3/1) and high (10/1) N/P ratios. The MTT results (Figure 5) revealed that most of the polymeric systems do not show signs of toxicity for Neuro-2a cells at the N/P ratios assayed. The cell viability was similar to that of control cultures in most vehicles, except for ECDP10 at both N/P ratios of 3/1 and 10/1 (viability < 75%, *p* < 0.001) and PEI25K at an N/P ratio of 75/1 (viability below 50%, *p* < 0.01). While the toxicity of PEI25K at high N/P ratios is to be expected [33], that of ECDP10 is surprising given that the incorporation of PEG is a widely used strategy to improve the biocompatibility of delivery systems [32], and this suggests that APEG concentrations over 5% give the systems toxic properties, which probably arise due to the strong interactions through hydrogen bonding between the hydroxyl groups of APEG and the ones of CD cavities (the higher the number of OH groups is, the better the interactions between them are).

### 3.5. Particle Size and ζ-Potential 

For these analyses, DNA oligonucleotides with a sequence equivalence to miR-138 mimic were complexed with polymers at an N/P ratio of 5/1. PEI25K polyplexes at an N/P ratio of 7.5/1 were included as references. The determination of the particle sizes of complexes was carried out using a size analyzer and Dynamic Light Scattering (DLS). The particle sizes shown in Figure 6A range between 100 and 270 nm. Complexes formed with the EPA homopolymer reached sizes of 225 nm, whereas PEI25K complexes reached sizes of 190 nm. EHPMA polyplexes are the smallest ones among the test systems, with sizes of 100 nm. The incorporation of EHPMA units may perform extra ionic interactions apart from contributing to the colloidal stabilization of polyplexes due to its amphiphilic character, allowing the formation of smaller particles than those built by the EPA homopolymer and other copolymer compositions [16]. Polyplexes formed by the ECD copolymer reached larger sizes (240 nm), which is probably due to hydrogen bonding between the hydroxyl groups of glucose units and aminos N or O or the carboxyl and phosphate groups of DNA, respectively. The incorporation of PEG into the ECD copolymer reduced the sizes of complexes, reaching 170 nm in ECDP5 complexes and 140 nm in ECDP10 complexes. APEG could act as a capping agent, progressively hindering particle growth with increasing concentration. 

The surface charge (ζ-potential) of complexes was assessed using a Zetamaster system. The surface charge of different polyplexes (Figure 6B) showed remarkable differences. Complexes formed by PEI25K, EHPMA, and EPA revealed positive ζ-potentials that indicate high particle stability in a solution (23, 27, and 12 mV, respectively). Complexes formed by the ECD copolymer also yielded positive ζ-potential, but with lower values (5 mV), whereas the incorporation of PEG into the ECD copolymer (ECDP5 and ECDP10) resulted in polyplexes with negative ζ-potentials (−7 mV in both cases). The charges of these oligonucleotide polyplexes are, in all cases, less positive than those in the plasmid DNA counterparts described by Redondo et al. [16,22], which may indicate possible difficulties to complexing small nucleic acids. However, the low positive charge of most polyplexes, or even the negative surface charge of ECD and ECDP, does not preclude cell binding. Indeed, our flow cytometry results (Figure 4) and microscopy analyses that followed indicate that ECD and ECDP polyplexes result in the highest cell binding values despite their negative surface charge, whereas EHPMA and EPA are less effective at inducing cell binding despite their highly positive charge. The copolymers’ microstructure would be more homogeneous when they are reacting with methacrylamides, as in the case of EPA and EHPMA, rather than in the case of methacrylamide EPA with the styrene group of HCD, which would lead to more heterogeneous materials; these are aspects that are to be reflected in their properties and in these kind of behaviors.

### 3.6. Microscopy Analysis of miRNA Cell Binding and Internalization into Neural and Neuronal Cells

To further explore the efficacy of selected polymeric vehicles, EPA, EHPMA, ECD, ECDP5, and ECDP10, as well as the reference polymer, PEI25K, to increase the rates of miRNA binding and internalization into neural cells, we performed microscopy analyses in Neuro-2a cells and rat primary hippocampal neurons after treatments with Cy3-labeled negative control mimic (Cy3 Neg. Ctrl mimic) complexed with different polymers at an N/P ratio of 5/1 (except for PEI25K that was complexed at an N/P of 7.5/1) and immunostained against ɑ-tubulin protein and DAPI nuclear dye. 

As shown in Figure 7, microscopy analysis of Neuro-2a cells confirmed the above results from the same cells via flow cytometry. PEI25K complexes contained about 70% miR+ cells. Compared to PEI25K, the EPA homopolymer resulted in significantly fewer miR+ cells (35% of miR+ cells, *p* < 0.01 relative to PEI25K). The incorporation of hydroxylated cyclodextrins to EPA, ECD, ECDP5, and ECDP10, increased the percentage of miR+ cells, resulting in more than 85% stained cells, without significant differences as compared to the proportion of those in PEI25K. On the contrary, the EHPMA copolymer led to significantly lower numbers of miR+ cells, even below the levels observed with polyplexes of EPA homopolymer (10% of miR+ cells, *p* < 0.01 respect PEI25K).

The results from the analysis of primary neurons were quite different. In these analyses, we estimated the percentage of neurons and of accompanying primary cells stained with Cy3 fluorophore of the Neg. Ctrl mimic, as well as the percentage of neurons in the culture. The results revealed that, on the one hand, the percentage of neurons in the culture strongly varied among treatments. The baseline (Cy3 Neg. Ctrl. mimic) cultures contained over a 50% neurons, and a similar value was observed after the administration of ECD polyplexes (53%), but these numbers dropped down to around 20% after the administration of PEI25K and of EHPMA polyplexes, whereas the EPA and ECDP polyplex cultures presented an intermediate percentage of neurons (aprox. 35%). Discrepancies in the percentage of neurons among treatments likely reflect differences in the neuronal toxicity of polyplexes, identifying that both PEI25K and EHPMA are highly toxic (*p* < 0.001). Indeed, after the administration of these polymers, the neurons did not develop neurites or show healthy morphologies such as those observed in the control cultures or after the treatment with ECD and derived copolymers (Figure 8A).

On the other hand, microscopy analysis also revealed that almost every polyplex leads to higher percentages of stained neurons, which are significantly higher than those in the control (Figure 8B). ECD polyplexes resulted in the highest percentages of stained neurons (68%, which corresponds to 33% out of 53% of the neurons in the culture), which are significantly higher than those of any other of the tested copolymers, including the reference, PEI25K, except for ECDP10. On the contrary, ECDPs were not able to reach the high values observed in Neuro-2a, with ECDP10 polyplexes leading to values of around 35% of the identified neurons (*p* < 0.001 relative to control), whereas ECDP5 yielded extremely low values (4%), which are equivalent to the negative controls (*p* > 0.05). The administration of EPA and EHPMA polyplexes resulted in intermediate percentages (34 and 29%, respectively), though they were significantly higher than those of the negative control (*p* < 0.01 and *p* < 0.05, respectively). PEI25K resulted in 27% miR+ neurons with the Cy3 Neg. Ctrl mimic (*p* < 0.05 respect the negative control).

The large cytoplasm of neurons allowed us to analyze the cellular distribution of miRNA staining to confirm whether the polyplexes became internalized or remained on the surface of the neuronal membrane. As Figure 8C illustrates, in all of the studied polyplexes, miRNA staining is observed on both at the cell boundaries and within the cytoplasm, indicating that different polyplexes are internalized in primary neurons.

Considered together, microscopy analysis revealed that the administration of ECD polyplexes resulted in the highest percentages of stained cells both among Neuro-2a cells and primary neurons. ECDP10 also resulted in high percentages of staining in both cells, but it exhibited some signs of toxicity for neurons, as observed for Neuro-2a in the toxicity assays. Neuro-2a and neurons also responded similarly to EPA and EHPMA polyplexes, in both cases with reduced percentages of miR+ cells; although, contrary to Neuro-2a, neurons showed potential signs of toxicity, particularly after the administration of EHPMA. On the contrary, PEI25K and ECDP5 showed the highest differences in the responses between neurons and Neuro-2a cells, with them being highly effective for Neuro-2a cells, but poorly effective for primary neurons, particularly ECDP5. The effects of polymers are clearly dependent on the nature of the cells under study, which should be taken into consideration in pre-clinical/clinical studies.

In light of all previous results, we decided to further analyze ECD and ECDP10 copolymers, studying their capabilities to facilitate the endosomal escape of the delivered miRNAs and the activity on their targets. In these analyses, we included PEI25K as a reference and the EPA homopolymer as a control, as well as the ECDP5 copolymer due to its good properties in all analyses despite its failure to deliver miRNA to neurons.

### 3.7. Endosomal Escape

Nanoparticles are typically taken up via endocytosis, and the endosomal/lysosomal compartments can be considered as the last biological barrier in RNA delivery [34]. In order to evaluate whether polymer vehicles induce cargo endosomal escape after cell internalization, we performed the calcein leakage assay described by Kongkatigumjorn et al. [27], which measures the escape of calcein from disrupted intracellular vesicles. To evaluate the ability of the polymer systems to produce endosomal escape, the C6 glioblastome cell line, which was chosen due to its large cytoplasm, was treated with calcein and different polyplexes, and 24 h later, the number of vesicles per cell was analyzed. As shown in Figure 9, C6 cells treated with calcein and negative control miRNA (Ctrl) showed a punctate distribution of fluorescence, which is indicative of dye storage within the endosomal/lysosomal compartments. This condition showed the maximum number of vesicles per cell. All polymer systems further decreased the number of vesicles per cell in comparison to that in the control condition (Ctrl), but the decreases were only statistically significant for the EPA homopolymer and hydroxylated cyclodextrin copolymers with APEG (ECDP5 and ECDP10), suggesting the polymer-induced escape of dye from endosomes. Although many studies described excellent endosomal escape effects of PEI [34,35,36,37], in our study, it did not significantly decrease the number of vesicles per cell in C6 cells, suggesting its poor capacity for the endosomal escape of the cargo, at least in this neural cell line. Moreover, our results showed that the incorporation of APEG into the cyclodextrin copolymers (ECDP5 and ECDP10) improved endosomal escape. These results differ from those in previous studies, which demonstrated that the addition of PEG inhibits cellular uptake and endosomal escape [38]. According to Moore et al. [39], the transfection efficiency of PEG-based vehicles that escaped the endosome was still limited by poor nuclear localization. Even so, many studies have demonstrated that an increased rate of endosomal escape is not directly related to an increase in transfection efficiency. It would be of interest in future studies to try and relate endosomal escape to the time and location at which it happens in the cell. This information could be relevant for avoiding premature degradation by the acidic environment of late endosomes or the drastic hydrolytic conditions of lysosomes [39,40].

### 3.8. miRNA Transfection Efficiency

To evaluate the functional cell transfection efficiency of different polymers, we performed dual-luciferase reporter assays in Neuro-2a cells using the pmiRGLO-cel-miR-67 plasmid in which luciferase expression is controlled by specific binding sites for cel-miR-67 miRNA and the cel-miR-67 negative control mimic (Neg. Ctrl mimic) complexed with the different polymers at N/P ratios of 5/1 and 10/1. 

Neg. Ctrl mimic transfection further decreased the level of luciferase activity when it was complexed with all polymer systems in comparison to that of the control cell culture (pmiR-cel-miR-67 transfected cells without Neg. Ctrl mimic co-transfection, dotted line in Figure 10). The decrease in luciferase signal was the largest (below the 60% of control signal, *p* < 0.01) after the transfection of Neg. Ctrl mimics with PEI25K at an N/P ratio of 7.5/1. Significant decreases were also observed in all other vehicles, though the decrease was lower. EPA, ECD, and ECDP10 polyplexes more significantly decreased the luciferase signal in comparison to that of the control signal when it was transfected at a low N/P ratio (5/1), with being ECD the most efficient polymeric system (30% signal reduction). However, at higher N/P ratios (10/1), all systems, except ECD, produced a significant decrease in the luciferase signal; although, none of them surpassed the 30% of signal reduction. Indeed, all EPA polymers lead to reductions in the luciferase signal, which were comparable to those of PEI25K, except ECD at N/P = 10/1. Despite these differences, it is interesting to note that the levels of the inhibition of reporter gene expression are similar to those obtained with PEI25K, whose action in miRNA delivery and silencing has previously been shown [41]. We observed that, unlike other polymers whose stable nanoparticle properties can be disadvantageous inside the cell [42], EPA and ECDP2 polymers allowed the cytosolic release of their payload to inhibit gene expression. Additionally, neither its short length nor its oligonucleotide backbone modifications had any significant negative effect on the functional carrier formation of the polyplexes assayed similarly to PEI F25-LMW [41].

## 4. Conclusions

RNA therapies are a promising field that has recently become a reality with efficient applications in fighting the COVID-19 pandemic. Despite their potential, delivery remains a major challenge, particularly for the nervous system and neural cells. In this study, we have synthesized and tested polymeric complexes based on EPA through all major steps of the nucleic acid delivery process. Our results suggest that some of the EPA-derived polymeric systems assayed here have properties, and these include: (i) they efficiently condense and protect the nucleic acid cargo from nuclease degradation; (ii) they facilitate the cellular uptake and endosomal escape into neural cells; (iii) they display little toxicity; (iv) they induce functional transfection of the miRNAs to a comparable level as the gold-standard polymer, PEI25K. Specifically, homopolymer poly-EPA, and its copolymers with β-cyclodextrins (ECD) with and without polyethylene glycol acrylate derivatives (APEG), show miRNA transfection efficiencies in the Neuro-2a cell line almost matching those of PEI25K, with minimal toxicity. Moreover, ECD and ECDP10 outperformed PEI25K in primary neuron transfection, even though functional analyses in neurons are required to confirm miRNA activity after delivery. On the contrary, copolymers with α-glucose-derived monomers, HPMA, and permethylated (MCD) β-cyclodextrins do not show appropriate carrier features, failing to fully complex miRNA mimics and to efficiently deliver them to neural cells, while some present toxicity issues at relevant N/P ratios.

The present results also demonstrate that the composition and microstructure of polymers, the nature of nucleic acid cargo, and the cells under study are determinants of carrier properties, including their nucleic acid condensation, polyplex size and charge, cytotoxicity, and transfection efficiency to neural cells and neuronal cells.

## Data Availability

Not applicable.

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
