# Peer review of "Evaluation of Poly(N-Ethyl Pyrrolidine Methacrylamide) (EPA) and Derivatives as Polymeric Vehicles for miRNA Delivery to Neural Cells"

_pharmaceutics, 2023, doi:10.3390/pharmaceutics15051451_

Round 1

Reviewer 1 Report

This manuscript synthesized series of Poly(N-ethyl pyrrolidine methacrylamide) (EPA) copolymers, and compared their miRNA transfection capability and efficacy in Neuro-2a cells and rat primary hippocampal neurons. However, the author did not elaborate on the selection of neural cells as the research object in the introduction, but gave up the fibroblasts cell lines that he investigated and found to have a good response to EPA. Moreover, a miRNA that is toxic to nerve cells was selected for delivery. What is the significance of this choice? Moreover, the experimental groups of the data in each part of the paper are not coherent, and it is difficult to compare the data before and after. I think the idea of this article itself is not clear enough to be published on pharmaceutics.

Author Response

However, the author did not elaborate on the selection of neural cells as the research object in the introduction, but gave up the fibroblasts cell lines that he investigated and found to have a good response to EPA

Answer: The central aim of the present study is to test whether poly-EPAs and their copolymers may be safe and effective carriers to deliver miRNAs (and siRNAs) to neural cells as they were to deliver plasmids to fibroblast cells. We agree with the reviewer that this central aim was not stated clearly enough in the submitted manuscript. To make this point more clear we have modified the introduction including:

  1. a brief introduction to RNAi therapeutics in the first paragraph: “Diseases ...” (lines 50-52; 54-55;58-62).
  2. a clarification of the precise aims and the design of the study in the last paragraph: “The present study ….” (lines 99-121).

Moreover, a miRNA that is toxic to nerve cells was selected for delivery. What is the significance of this choice?

Answer: as we have mentioned in the above commentary, we have chosen miRNAs as cargo because this is the central aim of the study, to evaluate the potential of these polymers to deliver miRNAs to neural cells.Regarding the toxicity of miRNAs, they are endogenous, non-toxic molecules, although they can activate cell death processes. This should not be the case neither for miR-138, which according to previous results is neuroprotective for Neuro2a and neurons (see Maza et al. 2022. DOI: 10.3390/biomedicines10071559, and references therein), nor for cel-miR-67, which is a C. elegans miRNA with no known targets in mammalian cells. In fact, we believe that the cell death values obtained from the flow cytometry analyses after treatment with the Cy3 negative control miRNA that suggest miRNA toxicity actually reflect the intrinsic and variable cell death that occurs in cell cultures. In fact, the treatment with Cy3 negative control miRNA with non-vehicle was included to set a baseline of toxicity to compare with the rest of the treatments. 

Moreover, the experimental groups of the data in each part of the paper are not coherent, and it is difficult to compare the data before and after

Answer: We acknowledge this pitfall in the original manuscript. To try to solve it we have added a broad description of the study design at the end of the introduction. The test indicates that: “The study was designed … … efficiency (lines 111-121).

I think the idea of this article itself is not clear enough to be published on pharmaceutics

Answer: As mentioned in point 1, we have tried to clarify the aim in the study in the introduction. We hope these modifications and those included throughout the rest of the text will make the idea and the results of this study clear.

Reviewer 2 Report

This manuscript demonstrated the potential of poly (N-ethyl pyrrolidine methacrylamide) (EPA) copolymers as miRNA delivery systems to neural cell lines and primary neuron cells. In this study, different copolymers were synthesized and evaluated through their size, charge, RNA condensation ability, cytotoxicity, cell binding, and endosomal escape capacity. Finally, the miRNA transfection capability and efficacy of EPA copolymers were evaluated in vitro. This work is well-designed in terms of copolymer synthesis, characterization, and in vitro transfection. The developed copolymers demonstrated low cytotoxicity, however, the miRNA transfection ability of which may not be ideal comparing to PEI. I suggest major revision of this manuscript, a few suggestions could be addressed by the authors:

1.      ‘RNAs were incubated 2.5 h at 65°C’ (line 133). What is the purpose of this procedure? Are RNAs stable in this condition? What is the correlations with the practical application?

2.      The synthesized polymers were used at N/P ratio of 5/1 for RNA delivery, but the PEI25K polyplexes at N/P ratio of 7.5/1 were included as reference (line 337). Please indicate the reason for choosing these different N/P ratios.

3.      In the miRNA transfection test (Figure 10), PEI25K showed superior efficiency than synthesized copolymers. Thereby, less amount of PEI25K could provide similar miRNA transfection compared to those synthesized copolymers. Thus, the author should test the toxicity of each carrier at the concentration in regard to equivalent transfection ability (Figure 5) instead of N/P ratio.

4.      Judging by the Cy3 signals in confocal images in Figure 8C, the synthesized copolymers did not show enhanced delivery efficiency of miRNA mimic than the control group.

5.      To support the neural cell miRNA transfection ability, the copolymer + miR-67/138 systems should be tested for more neural gene regulation rather than simply Luciferase activity.

Author Response

  1. Answer: The procedure described in line 151 is the standard protocol for the removal of the t-butyldimethylsilyl (TBDMS) group described by the group of Dr. Usman in 1995 (Wincott et al. 1995. Synthesis, deprotection, analysis and purification of RNA and ribozymes. Nucleic Acids Res 23, 2677-84.). This citation is a milestone in the preparation of RNA oligonucleotides. This research group was trying to synthesize long RNA ribozymes as potential therapeutic uses and they found that deprotection of TMDMS groups with tetrabutylammonium fluoride was inefficient for long oligonucleotides. They solved the problem using triethylamine hydrofluoride at 50-60 ºC in N-methylpyrrolidone solutions instead of tetrabutylammonium fluoride at room temperature. After this publication, all the nucleic acid chemists use Usman’s protocol to remove the TBDMS groups in RNA because it yields fully deprotected RNA in high yields. There is no degradation of the RNA. On the contrary, at room temperature, it is not possible to remove the TBDMS group, especially in long oligonucleotides. We have modified the text of the protocol to indicate that the treatment with triethylamine hydrofluoride at 65 ºC was done to remove the TBDMS group and add the citation of the protocol. The text now indicates:”... was added to remove the t-butyldimethylsilyl groups. TBDMS-protected RNAs were incubated 2.5 h at 65°C as described in Wincott et al [23]. The deprotection reaction was quenched and the crude product was purified by OPC cartridges (Glen Research). The oligoribonucleotides were quantified by UV absorption and characterized by MALDI-TOF mass spectrometry" (lines 159-163).
  2. Answer: We fully agree with the reviewer that this disparity can be confusing. N/P ratios of PEI25K polyplexes do not match the ratios of the other polymeric systems in the analyses because they were originally analyzed as mass/mass ratios and only later transformed into N/P ratios (N/P ratio=7.5 corresponds to a mass/mass ratio equal to 1). We employ m/m ratios for PEI25K because it was the routine procedure of the laboratory, particularly while trying to replicate the study by Hwang et al. (2011. Biomaterials 32(21), doi: 10.1016/j.biomaterials.2011.03.047) on the use of modified PEI vehicles for neuronal transfection. We have modified the text under the Polyplexes formation heading of the methods to explain and clarify this point (lines 201-211): "Polymers condense miRNA via ionic interactions of the amine groups (N) of the polymer and the phosphate groups (P) of the nucleic acid. Various formulations of polymer/RNA complexes (polyplexes) were prepared in distilled water or Tris-EDTA solution (TE, pH=7) depending on the assay with N/P molar ratios ranging from 1/1 to 100/1. Polyplexes of PEI25K at various N/P ratios were included as reference. N/P ratios of PEI25K polyplexes do not match the ratios of the other polymeric systems in the analyses because they were originally analysed as mass/mass ratios and only later transformed into N/P ratios (N/P ratio=7.5 corresponds to a mass/mass ratio equal to 1). For polyplexes formation, RNA and polymer solutions were combined and let stand for 20 min at RT". 
  3. Answer: We are not sure to fully understand the question raised by the reviewer. We completely agree with the reviewer in that PEI25K shows superior efficiency in the luciferase assay. We also agree with the reviewer in that this may indicate that reducing the amount of PEI can result in transfection efficiencies equivalent to the poly-EPA vehicles. However, PEI25K requires N/P ratios above 5/1 to reach its maximum complexation of miRNA mimics (Figure 2). This value, therefore, sets the minimum N/P ratio for transfection. Reducing below this level would result in increased amounts of uncomplexed miRNA mimics that will not be transfected to the cells. Concerning all other polymeric systems, transfection efficiency was measured within the N/P ranges (5/1 to 10/1) that were characterized for toxicity (3/1 to 10/1). Throughout this study, we have tested the carriers using the same parameters (avoiding particular bias) to select them according to their best assay performance.
  4. Answer: The purpose of the confocal images in Figure 8.C. is to evaluate whether the miRNA Cy3 is internalized within the cell or remains on its surface. As indicated by the reviewer, the signal of Cy3 negative control miRNA was observed within the cell after the administration of the miRNA, either free or complexed with the different polymers. This indicates that all treatments (including the free miRNA) were able to promote the internalization of the miRNA, not just to promote the binding to the cell membrane. However, we consider that the Cy3 signal within the cell mainly reflects the accumulation of miRNA within the endosomal-lysosomal compartment after internalization. If we were to analyze transfection efficiency we would need to measure the presence of active, free miRNA molecules within the cytoplasm following endosomal escape. Unfortunately, the employed microscopy techniques are not sensitive enough to visualize isolated miRNA molecules. Measuring overall cell fluorescence is too coarse to identify internalized miRNAs that appear free within the cytoplasm. Therefore, we consider that the best option to evaluate efficiency is to carry out a functional luciferase assay that reflects the presence and activity of miRNAs within the cell.
  5. Answer: We employ luciferase reporter assay because it is a standard assay used and widely accepted for functional validation of the activity of miRNAs. As the reviewer mentions, analyzing the expression levels from the targets of the transfected miRNA can be an alternative or complement to these analyses. However, cel-miR-67 has no described targets in mammalian cells and therefore, we cannot analyze their regulation. This limitation may have been overcome using miR-138 which has more than 250 experimentally validated targets according to miRTarBas (https://mirtarbase.cuhk.edu.cn/~miRTarBase/miRTarBase_2022/php/search.php) Unfortunately, in our experience, it can be difficult to employ the effects of miR-138 on its targets to measure transfection efficiency because: 1) the endogenous expression of this miRNA in mammal cells combines its effects with the transfected mimics; 2) the moderate effects of miRNA on target expression; 3) the existence of alternative regulatory pathways that can be also affected, particularly considering miRNAs targeting on hundreds of genes. Indeed, we opted for a Luciferase reporter assay based on cel-miR-67 to avoid all these limitations, using a miRNA without targets in the cells under study and a fully complementary regulatory sequence in the Luciferase construction that enhances the effects of the miRNA.

Reviewer 3 Report

Soto et al. synthesized ten different EPA copolymers and evaluated the effects of these polymers and EPA monomers as well as the transfection of gold standard PEI25k on miRNA protection, cytotoxicity, transfection effect, lysosome escape, etc. This work reveals the potential advantage of EPA copolymers in delivering miRNAs to neural cells. However, there are still some concerns as follows:

1.     Introduction section and abstract section should highlight the importance of the work and focus on explaining the logic of this work. That is, why this cell line was chosen for validation, and why these polymers were chosen for research, and what is the value of this work for future research in this field?

2.     The arrangement and legend of Figures need to be modified, some suggestions are as follows:

1)    It is not intuitive to use the table in Fig. 4A to present the results, so it is suggested to directly use the flow cytometry diagram to make it clearer. In Fig.4B, different colors may be used to distinguish, so as to improve the reading comfort.

2)    In Fig.5, it is suggested to use a bar graph instead of a dot graph.

3)    The legends are too long and not concise enough. An explanation of the results, as shown in Figure 3, can be included in the text or displayed directly in the figure.

4)    The red fluorescent background of some images in Figure 7A and Figure 8A is too strong, and it seems that it is not uniformly processed.

3.     It is suggested to polish the language of the manuscript, standardize the professional expression, and check the content of the manuscript throughout. Some formats and expressions need to be unified or modified, including but not limited to the following contents:

1)When the abbreviation appears for the first time, the full name should be attached, such as CNS;

2)In vitro and in vivo should be italics;

3)The number of the chemical formula should be subscript, For example, "CO2", "CDCl3" and "-CH2-O-CO -" in the legend of Figure 1;

4)Please use the same writing format for marking figure in the results and discussion section. As in "Particle sizes, shown in Figure 6.A., range between 100 and 270 nm" on page 15, "The surface charge of the different polyplexes (Figure 6.B) showed remarkable differences." on page 16, "As figure 8C illustrates" on page 18, etc.

4.     In the conclusion section, it is not easy to understand the causal relationship in the second paragraph, Why the transfection advantage of PEI is presented as a reason for the following content? The logic of this paragraph is a little confusing.

5.     Is it considered to be verified in animal experiments? Because the in vitro cell experiment may be unable to simulate BBB, resulting in a big difference from the in vivo delivery effect.

Author Response

  1. Answer: Thanks for the indication. We have tried to clarify all the points throughout the introduction. Please refer to the revised manuscript.
    1. Answer: Thanks for the comment. Flow cytometry experiments allowed us to acquire a large amount of valuable information that cannot be visualized simultaneously in a flow cytometry diagram. For this reason, and although we admit it is not the most intuitive way to do it, we consider that the table is the best way of summarizing our results without losing valuable information. In relation to the suggestion of using colors in Figure 4B, we are in accordance with the reviewer so we have included colored bars not only in this figure but also in all graphs to improve reading comfort. 
    2. Answer: Thanks for the suggestion. We have modified the figure representing data in a bar graph instead of a dot plot graph. Moreover, as commented above, we have colored the graph bars in order to improve reading comfort.
    3. Answer: Following the suggestion by the reviewer, we have reduced and harmonized all figure legends removing the explanation of the results.
    4. Answer: Thank you very much for the comment. It is true that the images included in the figure have not been processed in the same way regarding the green and red background. The reason is that Figures 7A and 8A include only one representative image per condition from all experiments performed, so these images do not necessarily come from the same experiment in all cases. These images were chosen because they were the best representing the mean (represented in each graph) of all experiments in each condition. However, within the same experiment replicate, the green and red fluorescent backgrounds were processed uniformly by using control conditions images as background reference. More in detail, images of control cells immunostained against É‘-tubulin (Neuro-2a cells) or ΙΙΙ−β-tubulin (primary hippocampal neurons) were used as reference to subtract red and green background signal from the studied conditions included in the figures.

    1. Answer: Done
    2. Answer: Done
    3. Answer: Done
    4. Answer: Done. All figures have been written with this format “Figure 4A”.
  2. Answer: We fully agree with the reviewer in the lack of clarity of these two sentences. We have decided to eliminate both sentences because they did not provide any fundamental information or conclusion. Moreover, sentences may be interpreted as a support to use these poly-EPA vehicles in vivo which is not our intention (see comment below), but to remark that these vehicles have the potential, after improvements, to be useful in vivo.
  3. Answer: Thanks for the suggestion. We fully agree with you. Indeed, this is our ultimate goal but, in our experience with the administration of miRNA mimics to the spinal cord, the polymeric systems as well as the lipofection agents that we have tested in vivo up to now –including PEI25K and related formulations– did not yield significant evidence of transfection. Therefore, we still need to improve the properties of the carriers, to confirm functional activity of the complexes in primary cultures of target neuron populations, and to take advantage of the rich pool of in vitro models for testing additional key aspects not analysed here (e.g. specificity), before going in vivo.

Round 2

Reviewer 1 Report

The authors have improved the manuscript and this revised version can be acceptable.

Reviewer 2 Report

The authors have addressed most of my concerns and now could be accepted for publication.

Reviewer 3 Report

The authors have improved the manuscript and this revised version can be acceptable.